# Immune–Pathological Correlates of Disease Severity in New-World Kala-Azar: The Role of Parasite Load and Cytokine Profiles

**DOI:** 10.3390/pathogens14070615

**Published:** 2025-06-20

**Authors:** Ingridi de Souza Sene, Dorcas Lamounier Costa, Daniele Alves Zacarias, Jailthon Carlos dos Santos, Gabriel Reis Ferreira, Daniela Rodrigues Andrade, Jorge Clarêncio de Sousa Andrade, Carlos Henrique Nery Costa

**Affiliations:** 1Laboratory of Leishmaniasis, Institute of Tropical Medicine Natan Portella, Teresina 64002-510, Brazil; ingssene@gmail.com (I.d.S.S.); dorcas.lc@gmail.com (D.L.C.); danizacarias_bio@hotmail.com (D.A.Z.); jailthonsilva@yahoo.com.br (J.C.d.S.); 2Maternal and Child Department, Federal University of Piauí, Teresina 64049-550, Brazil; 3Intelligence Center for Emerging and Tropical Diseases (CIATEN), Teresina 64049-550, Brazil; ferreira.rgabriel@gmail.com; 4Integrated Laboratory of Microbiology and Immunoregulation, CPqGM-Fiocruz-Bahia, Salvador 40296-710, Brazil; danirodriguesandrade@hotmail.com (D.R.A.); jorgec@bahia.fiocruz.br (J.C.d.S.A.); 5Department of Community Medicine, Federal University of Piauí, Teresina 64049-550, Brazil

**Keywords:** visceral leishmaniasis, *Leishmania infantum*, hemorrhage, sepsis, HIV-coinfection, mortality, bone marrow, innate immune response, adaptative immunity, IL-12

## Abstract

Kala-azar is a protracted disease caused by the protozoan *Leishmania infantum* (zoonotic) or *L. donovani* (anthroponotic), transmitted by sandflies. Patients present with fever, anemia, and hepatosplenomegaly, potentially progressing to hemorrhaging, secondary infections, and death. Its pathogenesis is linked to an exaggerated cytokine response. We studied 72 hospitalized patients, analyzing clinical data and outcomes in relation to *L. infantum* DNA loads in blood and bone marrow, and plasma concentrations of IL-1β, IL-6, IL-8, IL-10, IL-12, TNF-α, and TGF-β. Cytokine levels were found to be elevated. *L. infantum* kDNA loads in blood and bone marrow were strongly correlated and increased with disease duration. Higher parasite loads were observed in men, adults, and HIV-infected patients, and they were significantly associated with mortality. IL-6 was independently linked to sepsis. In multivariate analysis, IL-12 was the only cytokine inversely associated with blood parasite load. Parasite load, but not cytokine levels, correlated with disease severity, suggesting additional mechanisms drive progression. IL-12 appears to limit parasitemia, indicating a weak, persistent adaptive immune response that is ultimately overwhelmed by a progressive, inefficient, and inflammatory innate response.

## 1. Introduction

Kala-azar, or visceral leishmaniasis, is a lethal parasitic disease with protracted symptoms. Most patients report low-grade fever. Patients also report inappetence, diarrhea, and weight loss and appear malnourished and anemic, with an enlarged spleen and liver [1]. Jaundice, vomiting, and emaciation; signs of bleeding; and bacterial infections are frequent and may lead to death [2,3]. The disease is more common and more lethal at the extremes of age [4,5]. Immunosuppressed patients are also at higher risk [6,7]. 

Two species of protozoa are the cause of this disease. *Leishmania donovani* is the agent in East Africa and South Asia, and it is restricted to humans. Currently, *L. donovani* kala-azar has been brought under control in South Asia through a set of well-coordinated control measures [8]. *L. infantum* causes this disease among humans and other mammals in Central and Western Asia, the Mediterranean Basin, and the Americas. Unlike *L. donovani* kala-azar, the incidence of *L. infantum* kala-azar remains unaffected by control measures [9]. Transmission occurs mainly through being bitten by any of several species of sandflies. The flagellar promastigotes develop in the insects and culture media. After the infective bite, the promastigotes are phagocytized by neutrophils and monocytes. They lose their flagella and survive as amastigotes in macrophages distributed in the spleen, the liver Kupffer cells, bone marrow, and lymph nodes [10,11,12]. They circulate in the blood inside monocytes and neutrophils in very low concentrations [13,14]. 

There are no known *Leishmania* virulence factors that can directly harm mammal host cells or tissues and cause disease [3]. Instead, they lead to sickness through host responses as long as they surpass the host’s innate and acquired defenses and progressively multiply, generating the typical signs and symptoms and, eventually, death [15]. As with other infectious diseases, systemic inflammation is triggered by pro-inflammatory cytokines, while regulatory cytokines limit inflammation and presumably immunity [11,16]. This wave of cytokines is associated with hemorrhagic manifestations and bacterial infections [17]. 

How precisely the parasites and host intertwine such that they develop the lethal disease phenotype has been investigated but is still a matter of conjecture [18]. Evaluations of the role of the host showed that cytokines such as IL-6, IL-8, IFN-γ, and sCD14 might be involved in more life-threatening disease [17,19]. Similarly, the larger the *L. infantum* load, the more severe the disease [20,21]. Interestingly, the *L. infantum* genome explains around 80% of the mortality of human patients, suggesting a complex interaction between the parasite and the host response [22]. However, it is not known how the interplay of parasites and host molecules leads to death even after prompt diagnosis and treatment. Mortality has remained 10% over the years, and it is increasing in some parts of the world [23]. Therefore, this study was developed to shed light on the connections between *L. infantum* burden, a set of host molecules, life-threatening kala-azar, and immunity in order to better understand the pathogenesis of severe kala-azar.

## 2. Materials and Methods

### 2.1. Patients

All kala-azar patients were treated at the “Natan Portella” Tropical Disease Institute in Teresina, Brazil, and kala-azar cases confirmed through the presence of *L. infantum* amastigotes on bone marrow smears or the presence of promastigotes in culture were included in this study. This study was performed on a sample of 72 patients, sequentially admitted with available clinical information obtained by a single physician and with cryopreserved plasma, blood, and bone marrow samples obtained before treatment.

### 2.2. Medical Data

A detailed clinical history and physical examination—including information on hemorrhagic manifestations—were conducted upon admission for all hospitalized participants. The recorded bleeding variables included any history of hemorrhagic manifestations either upon admission or during a hospital stay—such as gingival or gastrointestinal bleeding, hematoma at the venipuncture site, petechiae, and bruising. These instances were categorized into two variables: ‘reported bleeding’, if not directly observed by the medical team, and ‘detected bleeding’, if clinically observed. The former was based on patient-reported symptoms, while the latter was identified by medical staff. A patient was considered to have a bacterial infection if signs or symptoms were present and if the infection was confirmed through X-rays, urinalysis, or culture results. Infections were categorized into two variables: ‘sepsis’ and ‘any bacterial infection’. Sepsis was defined by the presentation of hyperventilation, tissue hypoperfusion, decreased venous oxygen saturation, oliguria, or altered consciousness. Accordingly, the variable ‘sepsis’ referred to patients who met these criteria, while ‘any bacterial infection’ included all diagnosed bacterial infections, regardless of severity.

Since the number of deaths was too small for detecting statistically significant clinical associations with severe disease, the score system software Kala-Cal^®^ was also used as a proxy for disease gravity. This system uses data such as edema, jaundice, dyspnea, HIV coinfection, vomiting, bacterial infections, and hemorrhages, and, indeed, its results showed significant associations with mortality. Data on this software have been published elsewhere [5], and this program can easily be accessed at https://www.sbmt.org.br/kalacal/ (accessed on 17 March 2024). In synthesis, disease severity was evaluated by six variables: “death”, “chance of death > 10% by using Kala-Cal^®^”, “reported bleeding”, “detected bleeding”, “sepsis”, and “any bacterial infection”.

### 2.3. DNA Isolation, Purity, and Standardization

Bone marrow and blood samples were collected in both heparinized and citrated tubes. Isolated parasites were stored in liquid nitrogen. DNA isolation was performed using the QIAmp DNA Blood Mini Kit (Hilden, Germany), according to the manufacturer’s instructions, with 200 μL of plasma or bone marrow. The purity and DNA concentration were evaluated using a spectrophotometer (NanoDrop ND-1000 spectrophotometer; Thermo Fisher Scientific, Waltham, MA, USA). DNA samples were normalized to a concentration of 5 ng/μL and concentrated or diluted with distilled water when necessary.

### 2.4. Quantitative PCR

Quantitative PCR was carried out by using the TaqMan probe to quantify *L. infantum* in plasma and bone marrow accurately. The target sequence for *L. infantum* detection consisted of FAM–TTT TGA ACG GGA TTT CTG-MGB-NFQ (GenBank AF169140). Specific primers based on kinetoplast DNA consisting of 5′–GGC GTT CTG CAA AAT CGG AAA A–3′ (forward) and 5′–CCG ATT TTT GGC ATT TTT GGT CGA T–3′ (reverse), (Applied Biosystems, Foster City, CA, USA) were used [21]. A standard curve was constructed using 10-fold serially diluted *L. infantum* DNA corresponding to 10^4^ to 1 parasite per reaction.

Albumin was selected as the housekeeping gene to equalize the parasite count in the bone marrow. The number of parasites was expressed as a ratio with respect to the number of human nucleated cells. The primers were 5′-GCT GTC ATC TCT TGT GGG CTG T–3′(forward) and 5′–ACT CAT GGG AGC TGC TGG TTC–3′ (reverse). The probe was VIC-GG AGA GAT TTG TGT GGG CAT GAC A–TAMRA (GenBank NG009291) [24]. A standard curve was constructed using 10-fold serially diluted human cells DNA corresponding to 2 × 10^4^ to 2 human nucleated cells per reaction.

Amplification and detection were performed using the StepOne™ Real-Time PCR System (Applied Biosystems, Foster City, CA, USA). Cycling parameters were 50 °C for 2 min, 95 °C for 10 min, and 40 cycles at 95 °C for 15 s and 60 °C for 1 min. Standards, samples, and negative controls were analyzed in duplicates. The threshold cycle (Ct) value was plotted in accordance with the standard curve. The cut-off between reactions was 20%, and no deviation proportions were considered.

### 2.5. Plasma Cytokines

The plasma specimens were stored at −20 °C. Serum IL-1β, IL-6, IL-8, IL-10, TNF-α, and IL-12 levels were measured using a high-sensitivity multiplex inflammatory cytokine panel via cytometric bead array (CBA); we also measured the levels of the cytokines IL-17 and TGF-β, which were measured independently using CBA-flex set (BD Biosciences, San Jose, CA, USA) on the BDFAcs Array (BD Biosciences, San Jose, CA, USA), following the manufacturer’s instructions. Measurements of each sample were performed in duplicate, and the average of the two measurements was used. Standard curves were derived from the cytokine standards supplied with the kit and subjected to 10-fold dilution. The lower limits of detection for specific analyses ranged from 0.40 pg/mL for IL-8 to 0.04 pg/mL for IL-1β, 0.04 pg/mL for IL-6, 0.97 pg/mL for IL-10, 0.03 pg/mL for TNF-α, 0.06 pg/mL for IL-12, and 0.05 pg/mL for TGF-β, based on standard-curve dilution. 

### 2.6. Statistical Analysis

Proportions and 95% confidence intervals were calculated for the clinical and demographic variables. The median and interquartile intervals, as well as the means, were calculated for kDNA loads. The Kolmogorov–Smirnov test was used to compare the plasma cytokines with the standard values taken from the literature [25]. The Wilcoxon rank-sum test was used to compare plasma and bone marrow kDNA loads according to demographic and clinical data. Similarly, this test was used to analyze the concentration of cytokines. Pearson’s correlation test was used to evaluate the correlation between death, “risk of death > 10% by Kala-Cal^®^”, “reported bleeding”, “detected bleeding”, “sepsis”, and “any bacterial infection”. The Spearman’s correlation test was used to test the correlation between the natural logarithm of plasma and bone marrow kDNA burdens and their correlation with the time of disease, as indicated by the time with fever. Finally, simple and multiple linear and quantile regression were applied to test whether the cytokines predicted the plasma and bone marrow kDNA loads. The statistical package Stata/IC 15.1 (College Station, TX, USA) was used to analyze the data. 

## 3. Results

### 3.1. Study Population

Table 1 shows the characteristics of the study population. Forty-two of the patients were male (58.3%), and thirty were female (41.7%). The median age was 7.5, and the mean was 15.2 years. Seventeen participants (23.6%) were under two, 22 (30.6%) were under four, and 61 were children under 15. Six were older than 40 (8.3%). Among the 70 who were tested for HIV, 13 were positive (18.6%). Four (5.6%) patients died. Male sex was associated with HIV infection (*p* < 0.05).

### 3.2. Clinical Findings

Table 1 also shows the clinical characteristics of the study population. The mean Kala-Cal^®^ chance of death was 12.1%, and the median was 6.5. Twenty-five patients had a chance of death of at least 10% (34.7%). More than 40% had hemorrhages or infections. Fifteen patients had a “reported bleeding” manifestation (20.8%). Ten patients fulfilled the criteria for sepsis (14.1%). Four patients had some sort of bleeding during their hospital stay, e.g., “detected bleeding” (5.6%). A bacterial infection of any type was detected in 23 (31.9%) patients. Mortality was associated with “detected bleeding” (*p* < 0.05) and sepsis syndrome (*p* < 0.01), as well as with any bacterial infection (*p* < 0.10). The six markers of disease severity were significantly and positively correlated, except “death” plus “reported bleeding” and “probability of death > 10%” plus “sepsis” or “any bacterial infection”, which were positively but not statistically correlated. The Appendix A provides information on the correlation matrix between these variables. 

The chance of death calculated using Kala-Cal^®^ was well correlated with the occurrence of death (r = 0.40, *p* < 0.001). As expected upon admission, the estimated chance of death among those who survived was 8.9%, much less than the chance for those who died later (65.3%). 

### 3.3. Quantity of Parasite Load and Cytokines

Table 2 depicts the plasma and bone marrow *L. infantum* loads, plasma cytokine concentrations, and reference values. The median plasma kDNA concentration was 856.7 kDNA amastigote equivalents/mL (AEq/mL), and the mean was 3515.4 AEq/mL. The median bone marrow kDNA concentration was 55.7 AEq/10^6^ DNA equivalents of human cells (HCEq), with an interquartile interval of 3.7–400.7 AEq/10^6^HCEq. The standard deviation was 1.7 times greater than the mean plasma kDNA, while the metric for the same relationship was 3.2 times greater for bone marrow kDNA, demonstrating the higher variability of the bone marrow count. The standard values were assumed to be zero, although some asymptomatic patients may have harbored minimal plasma kDNA loads [26,27,28,29]. Regarding the cytokines, IL-6, IL-8, IL-10, and TGF-β had medians and means much higher than the values of a healthy population, while IL-1β and IL-12 exhibited a more modest increase. Despite the high plasma concentrations, cytokines were not detected in a relevant proportion of patients: 28% for IL-12, 19% for IL-1β, 9% for TNF-α, and 6% for IL-6.

### 3.4. Plasma and Bone Marrow L. infantum Loads

The Spearman’s correlation test revealed a moderate but statistically significant correlation between the natural logarithms of the plasma and bone marrow parasite loads (r = 0.48, *p* < 0.001) (Figure 1). Bone marrow kDNA load without including human cell counts as the denominator in the parameter was poorly correlated with the parasite plasma load (r = 0.21, *p* > 0.05).

### 3.5. Time of Disease, Plasma and Bone Marrow L. infantum Loads, Plasma Cytokines, and Severity

The plasma and bone marrow *L. infantum* kDNA loads increased with the duration of disease, as estimated by the duration of fever (Figure 2). The Spearman’s correlation coefficient for plasma kDNA over time was 0.33 (*p* < 0.005), and for bone marrow, it was 0.38 (*p* < 0.05). There was no correlation of any cytokine with time. According to the univariate linear regression analysis of the relationship between the natural logarithm of time with a fever and the natural logarithm of plasma kDNA load, the probability of rejecting the model was <0.005, and the *p*-value of the coefficient was 0.005. For predicting the natural logarithm of the bone marrow load, the probability of rejecting the model was <0.001, and the *p*-value of the coefficient was <0.001. There was no association between any cytokine and the duration of disease. A greater chance of death and “detected bleeding” were slightly correlated with the duration of a fever (ρ = 0.23, *p* > 0.05 and ρ = 0.24, *p* < 0.05, respectively).

### 3.6. Plasma and Bone Marrow L. infantum Loads, Age, Sex, HIV Infection, and Kala-Azar Severity

Table 3 shows the relationships between *L. infantum* load and age, sex, HIV infection, and markers of kala-azar severity. Individuals 15 years of age or older had higher plasma median loads (*p* < 0.05), and patients 40 years old or older had the highest plasma loads (median = 9129.5 AEq/mL). The same trend was observed for bone marrow (*p* < 0.05). Plasma and bone marrow loads were higher in men (*p* < 0.05 and *p* > 0.05, respectively). HIV-infected patients had eight-times-higher plasma loads than non-infected patients (*p* < 0.01). On the other hand, in bone marrow, although the loads were higher, the difference was non-significant. In the multivariate quantile regression, HIV status did not stand out, and only sex and age were associated with plasma kDNA load. 

The four patients who died had kDNA loads in their plasma that were a median of 14.2 times higher than those in the survivors but non-significantly associated with death, likely due to the small number of deceased patients. The bone marrow *L. infantum* loads were also higher in the deceased patients in comparison to those in the patients who survived, but not in a statistically significant manner. The concentration of kDNA in the plasma was 11.4 times higher in those with a chance of death estimated to be above 10% (*p* < 0.001). In the bone marrow, this concentration was 14.5 times higher in those with a chance of death greater than 10% (*p* < 0.005).

The associations for the two variables related to hemorrhagic phenomena, “reported bleeding” and “detected bleeding”, were not concordant regarding *L. infantum* loads in the blood and bone marrow. Plasma kDNA loads were ten times higher in the four patients with “detected bleeding” than in those with “reported bleeding”. The opposite was observed for bone marrow: the *L. infantum* load was not very high in those with “detected bleeding”, but it was 10 times higher in the patients with “reported bleeding”. 

### 3.7. Plasma Cytokines, Age, Sex, HIV Infection, and Markers of Kala-Azar Severity

Table 4 shows the associations between age, sex, HIV infection, and the clinical manifestations of the severity of kala-azar with respect to the seven measured plasma cytokines. There was a paucity of statistically solid associations between plasma cytokines and the demographic and clinical data. IL-10 levels were almost two times higher in children (*p-*value < 0.05). IL-10 levels were also much higher in women than in men (*p*-value > 0.05). Similarly, IL-10 was the only cytokine associated with HIV infection, with its levels being significantly lower in those with HIV (*p*-value < 0.05). Possibly due to the small number of deaths in the study population (four), no cytokine had any statistically significant association with death. However, the median IL-6 level was noticeably, i.e., more than two times, higher in those who died. The proxy for death and disease severity used, i.e., the chance of death according to Kala-Cal^®^, showed no significant association with any cytokines. Only three patients with measured cytokines had “detected bleeding”. However, in these three patients, IL-6 levels were almost four times higher (*p* < 0.05). Only IL-8 was associated with “reported bleeding” (*p-*value < 0.05). Sepsis syndrome had some significant associations with cytokines. IL-12 levels were higher in those with the syndrome (*p* < 0.10), and the increases were nearly significant for IL-1β (*p* < 0.10), IL-6 (*p* < 0.10), and IL-10 (*p-*value > 0.05). After obtaining nearly all of the results, we found that the variable “any bacterial infection” was statistically associated with IL-1β (*p*-value < 0.05), IL-6 (*p*-value < 0.05), IL-8 (*p-*value < 0.05), IL-10 (*p-*value < 0.05), and TNF-*a* (*p*-value < 0.05). However, no statistically significant association would exist if the Bonferroni analysis for multiple comparisons were applied, that is, at a level of significance of the *p*-*value* < 0.001. In the multivariate linear regression, IL-6 was the only predictor of sepsis (*p-*value < 0.05), as confirmed in a multiple quantile regression. No cytokine was found to be able to predict the chance of death or any other clinical outcome at the *p* < 0.05 level.

### 3.8. Regression Analysis Between L. infantum Load and Plasma Cytokines

In the univariate linear regression analysis, IL-12 was found to negatively and significantly predict plasma kDNA levels (*p* < 0.01). Similarly, TNF-α also yielded negative and significant predictions (*p* < 0.05). However, when controlling for IL-12, the effect of TNF-α was no longer significant. In the multivariate linear regression, only IL-12 remained a significant predictor of plasma kDNA load (Appendix A); no other cytokines were associated with this factor. Finally, no cytokines predicted kDNA levels in the bone marrow. The models’ adjusted R^2^ and pseudo R^2^, respectively, were low, indicating that the studied cytokines were poor predictors of *L. infantum* load.

## 4. Discussion

This study found that patients with kala-azar have a high inflammatory status and show signs of a substantial risk of death. On the other hand, the data also shows that children, women, and persons living with HIV have a more prominent regulatory immune response. Markedly, however, IL-1β, IL-6, IL-8, IL-10, IL-12, TNF-α, and TGF-β were not or only poorly associated with the risk of death. Moreover, while the levels of the studied cytokines did not increase with the duration of the disease, plasma and bone marrow *L. infantum* loads increased progressively. Additionally, plasma and bone marrow *L. infantum* loads were highly correlated, and parasite loads were higher among men, adults, and HIV-infected patients. Finally, higher plasma and bone marrow kDNA loads were associated with increased severity. Most plasma cytokines were not related to plasma or bone marrow *L. infantum* load, but higher plasma IL-12 levels were independently associated with lower *L. infantum* loads.

This study’s weakest point is its cross-sectional design, which did not allow the identification of causal relationships, except with death. Other key cytokines and molecules, such as IFN-γ, were not studied. Additionally, the levels of bone marrow cytokines were not measured, and spleen size was not regularly registered, hiding this organ’s importance in regard to the observed values. However, the findings are of interest for helping us to understand the pathogenesis of life-threatening kala-azar and highlighting the progressive immunological failure experienced by patients with kala-azar.

This study population reflects the well-known male and child predominance with respect to kala-azar caused by *L. infantum* [30]. The degree of HIV coinfection was higher than usual for this region and country [31,32]. Although mortality was lower in this sample, more than one-third of the patients had a chance of death estimated to be over 10% and hemorrhages and bacterial infections. These findings demonstrate how severe the disease can be and stress the importance of understanding the pathogenetic mechanisms of these complications. When compared with levels in blood donors [25], the plasma cytokine concentrations in the patients were very high, confirming the role of inflammation. Indeed, previous studies have already shown very high values of the cytokines IFN-γ, IL-8, TGF-β, L-10, and IL-6 in the plasma of patients from East Africa, Brazil, and India [33,34,35,36].

Children had higher concentrations of most cytokines than older patients, but only the levels of IL-10 reached statistical significance. This finding might support a previous observation and suggests children have a more immunotolerant state, independently of kala-azar infection [37]. Regarding sex, IL-10 levels were higher in females. Although another study on healthy adult individuals did not observe this difference, the present findings suggest that a sex-dependent IL-10 response in patients with kala-azar may actually exist [38]. 

Another previous study showed that HIV-infected patients undergoing antiretroviral therapy (ART) had lower IL-10 levels than both patients not on ART and long-term non-progressors [39,40]. In the present study, IL-10 levels were found to be lower in patients coinfected with HIV than in those non-coinfected, suggesting that HIV infection attenuates the regulatory innate response to kala-azar. 

Based on observational data, it has been proposed that plasma IL-6 and IL-8, IFN-γ, IL-27, and soluble CD14 are the major mediators of the pathogenicity of kala-azar [3,17,19]. Due to its overlapping activity with respect to kala-azar, such as hemorrhages, anemia, hypoalbuminemia, and hyperglobulinemia, IL-6 was proposed to be the best explanatory cytokine for the complications of kala-azar [41]. However, in the present study, neither the chance of death upon hospital admission nor the occurrence of death revealed sufficiently strong associations between plasma cytokines and signs of severity. Indeed, when the *p*-values were corrected using the Bonferroni method for multiple comparisons, no association between clinical presentation and any of the seven studied cytokines was found. Additionally, in the multivariate linear regression, no cytokine was found to predict the chance of death. This set of data on the role of cytokines detected in the blood suggests that the core of the pathogenic phenomena that lead to severe kala-azar or death may not rely primarily on the direct effect of pro- or anti-inflammatory cytokines, as previously proposed [17,19]. 

In this study, we observed that the cytokine response does not change with time after the disease starts, and this finding may influence our comprehension of the framework for the pathogenesis of kala-azar. Although cytokine levels did not change, *L. infantum* loads and disease severity increased with time. This phenomenon does not seem to be determined by progressive spleen enlargement since parasite loads in the bone marrow also increased similarly. 

The median plasma and bone marrow amounts of *L. infantum* kDNA in this sample were in the range of the concentrations previously found in Teresina and elsewhere using the same protocol [20,21,42]. As expected, plasma and bone marrow kDNA concentrations were highly correlated, indicating that bone marrow is balanced with systemic parasitism. Men and adults in general had higher plasma and bone marrow loads, likely due to the modulatory effects of testosterone and dihydrotestosterone on men, who constitute the majority of adult patients with kala-azar [43]. Interestingly, HIV infection was not independently associated with a higher parasite load. One possible explanation for this is that the patients with HIV had a relapsing course of kala-azar and, therefore, were undergoing secondary prophylaxis with liposomal amphotericin B for *L. infantum* to prevent kala-azar relapses. 

The disease severity associated with *L. infantum* load is a relevant finding and deserves further discussion. Unfortunately, this study design does not allow for the assessment of the direction of causality, i.e., if *L. infantum* load worsens the disease via a linkage with a specific, unknown factor or if a broad, ongoing multifactorial lymphoid disruption leads to a non-specific, progressive, and generalized failure of immunity and then a higher parasite load. 

Cytokines’ direct contributions to the manifestations of severe disease have already been studied [17,19], but here, plasma cytokines were also compared with plasma and bone marrow *L. infantum* kDNA loads, clinical presentation, and risk of death. IL-12 and TNF-α were found to be associated with plasma *L. infantum* load, but IL-12 was the only one found to be an independent predictor of plasma kDNA load, and, importantly, this interaction decreased *L. infantum* loads in a dose-dependent manner. Indeed, this finding is consistent with the canonical role of innate-immunity-derived IL-12, which promotes antigen-specific Th1 responses via T-cell activation, proliferation, and differentiation through the secretion of IFN-γ [12,44,45,46,47,48,49]. Unfortunately, this finding has not been forecasted, and IFN-γ was not analyzed in the present study. However, two previous articles analyzed the association between plasma cytokines and the blood load of *L. donovani* kala-azar in India and Africa [24,50]. They found a positive correlation with IL-10, TGF-β, and IL-17 but not with IFN-γ, TNF-α, IL-6, IL-4, IL-2, IL-12, and IL-22, suggesting that immunoregulatory cytokines are the primary controllers of blood parasite loads in *L. donovani*-derived kala-azar. However, although Teles et al. [51], in Brazil, and van Dijk et al. [36], in Uganda, also found a positive correlation with IL-10, they identified a negative correlation with IFN-γ. The difference between the data from India and the findings in Brazil and Uganda suggests that *L. infantum* and Indian *L. donovani* differ in terms of the host control of kala-azar: while there seems to be a more prominent role of a sustained, acquired, type Th1 response for the disease caused by *L. infantum* and Ugandan parasites, in the disease caused by Indian *L. donovani*, innate regulatory cytokines “deal the cards” and have an absolute, permissive effect on parasite load.

IL-12/IL-10 interaction is the line of balance in kala-azar: at the infection site, intracellular amastigote molecules drive the activity of the infected macrophages to promote a predominant IL-12/ IFN-γ or IL-10/IL-27 synthesis [10,15,52]. In the majority of infections, a Th1 response prevails, stimulating T-cells to synthesize IFN-γ, which triggers macrophages to produce free radicals that kill the amastigotes, and thus the infection is controlled [11,12,53,54]. In a small proportion of immunocompetent humans, IL-10/IL-27 prevails and disrupts macrophage IL-12 signaling to CD4+ and CD8+, blocking the secretion of IFN-γ and preventing macrophage activation for intracellular defense [48,54,55,56,57,58]. All these events may happen entirely at the innate-immunity level. However, with time, acquired immunity develops. If there is a stronger Th1-type response with T-cells secreting IFN-γ, memory T-lymphocytes generate *Leishmania*-specific clones, and the host becomes immune, as indicated by the high proportion of persons with cellular immunity to *Leishmania* who never develop symptoms and live in endemic areas [26,59]. However, if a regulatory profile is maintained, parasite load increases, disease and complications appear, and the host eventually dies. This study shows that despite the dominance of IL-10’s effects on *L. donovani* kala-azar, acquired immunity persistently influences parasitism in *L. infantum*-derived and Ugandan kala-azar.

Therefore, the IL-12 response in *L. infantum*-derived kala-azar reported here and the IFN-γ response reported elsewhere suggest that an overwhelmed but enduring acquired immunity persists for a while during the course of this disease. Another reason for this hypothesis is that the secretion of IL-12 is maintained for a longer period after the disease is cured in comparison to what is observed for the cytokines secreted after the stimulation of the innate response, such as the rapid decrease—at one to two weeks—in the levels of the cytokines mentioned above, as previously reported [36,60]. Therefore, IL-12 seems to be part of acquired immunity, not innate immunity, since it is long-lasting due to the memory T-cells developed after earlier antigenic priming of T-cells, while IL-1, IL-6, IL-8, IL-10, and TNF-α, albeit at higher concentrations, last only during the antigenic stimulus, as characteristically occurs with respect to innate immunity. Nevertheless, although IL-10 has been described to be associated with T-regulatory cytokines as part of acquired immunity, it typically falls to very low levels after kala-azar is cured [34,35,36,61].

Similarly, the early decrease in IFN-γ levels observed in the study by Lima et al. [60] and van Dijk et al. [36] suggests that most plasma IFN-γ originates from cells that are part of the innate immune system, such as neutrophils, eosinophils, NK cells, or even T-cells, in an antigen-independent process [34,62,63], not from TCR antigen-specific CD4+ or CD8+ T-cells. Therefore, this study suggests that in kala-azar, innate and acquired immunity coexist with the disease. With treatment, the hidden cellular immunity is established, but relapses indicate that the parasite persists even after immunity develops and patients become asymptomatic, e.g., non-sterile immunes. This equilibrium may have advantages for both: long-lasting immunity at the cost of some chance of disease remission and transmission. 

While *L. infantum*-derived kala-azar still shows signs of some effective defense, *L. donovani*-derived kala-azar does not. The two parasites lead to remarkable phenotypic differences, despite being relatively genetically similar. *L. infantum*-based kala-azar seems to derive from the older *L. donovani* kala-azar [64]. While *L. donovani* kala-azar is mostly transmitted among humans, *L. infantum* kala-azar is zoonotic, with a breadth of mammal hosts [1]. *L. donovani* kala-azar develops in older individuals, while *L. infantum* kala-azar affects younger, immunocompetent persons, primarily children, indicating a higher force of infection [30]. *L. donovani* kala-azar leads to post-kala-azar dermal leishmaniasis, which is rare in infections with *L. infantum* [65]. There are other clinical and epidemiological differences between the two species, but comparative, well-controlled, head-to-head studies on these two species and places still need to be performed, covering topics ranging from genomic analysis to innate and acquired immunity and pathogenesis.

Another open question from this study is why bone marrow *L. infantum* load did not show an association with IL-12 or TNF-a similar to that shown by plasma load. One explanation is that bone marrow control of *Leishmania* parasitism has in situ peculiarities not captured by the plasma cytokines analyzed. An insightful study investigated the association between bone marrow cytokines and local *L. infantum* burden. The findings revealed that IFN-γ was associated with a reduction in bone marrow parasite loads, whereas interleukin-10 (IL-10) correlated with an increase in parasite burden. Interestingly, a strong and statistically significant **positive correlation** between IFN-γ and IL-10 levels was also observed. However, IL-12 or TNF-a were not found to be associated with parasite burden [51]. These findings illustrate the strongly opposing effects of concomitant and protective versus permissive cytokines, each one downregulating (or upregulating) the other, in the bone marrow. It was not clear if the local findings in bone marrow can be generalized for blood.

Since cytokines are pleiotropic and redundant, engaging in synergistic actions, it is difficult to understand their individual roles in diseases [57]. Here, the risks of death and complications were not firmly associated with specific cytokines, except the association of IL-6 with sepsis in the multivariate regression analysis. However, since sepsis is the main cause of cytokine storms [57,66], it is not valid to infer that the increase in IL-6 levels was due to *L. infantum* instead of lipopolysaccharide (LPS) from opportunistic bacteria. Remarkably, mortality and the chance of death were not associated with any plasma cytokines. Although IFN-γ was not analyzed in this study, it does not seem to be a candidate for explaining kala-azar severity since its side effects are mild in a human host and do not match the symptoms of complicated kala-azar [67].

Therefore, the important question is how a microorganism without virulence factors that do not directly harm human cells or tissues could lead to death [15,18]. A progressive rise in parasite load indicates a failing acquired immunity, suggesting that a higher parasite load is a consequence and not a cause of immune failure. Two distinct mechanisms without the direct and specific action of cytokines may explain the global defense failure. One is T-cell exhaustion, and the other is spleen disorganization. T-cell exhaustion is a dysfunction of T-cells, mostly CD8+ cells, occurring naturally and during chronic infections and cancer [68]. The other alternative is the disorganization of lymph nodes and spleen architecture, a phenomenon that disrupts the white pulp structure [69]. Both mechanisms are exacerbated in kala-azar infections and may hypothetically result in an increasing amastigote load, progressive acquired immune failure, and, thereafter, an increased probability of complications and death [70,71,72]. These two mechanisms of immune deterioration occur simultaneously in kala-azar, but a clear molecular link between them has not been identified yet [73]. Hence, these processes seem to be accelerated in kala-azar by an elusive systemic factor, likely linked to the prolonged and intense inflammatory status; lymphotoxin-β and IL-27 are plausible mediators of this phenomenon [74,75].

In summary, *L. infantum* load was associated with life-threatening kala-azar, but the corresponding mechanism is unknown. In contrast, circulating cytokines were poorly associated with phenotypes of severe disease. As expected, IL-12 has an enduring, strong, negative effect on *L. infantum* proliferation during kala-azar infection. Parasite load worsened with time, but cytokine load did not, suggesting a cytokine-independent immunological failure process that results in progressively severe disease and death. However, there are still no key host factors leading to complications and death by kala-azar. Consequently, it may only be conjectured that the long-term immunological consequences of sustained infection and inflammation, such as persistent immune activation, may lead to immune exhaustion and overall immunological disorganization. 

## Figures and Tables

**Figure 1 pathogens-14-00615-f001:**
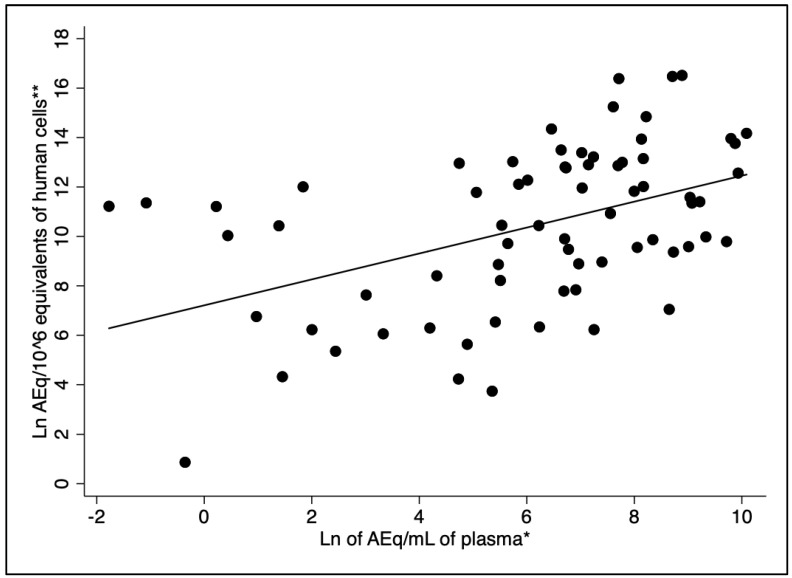
Correlation between plasma and bone marrow parasite load (r = 0.48, *p*-value < 0.001). Legend: * Natural logarithm of kDNA amastigote-equivalents/mL of plasma (AEq/mL). ** Natural logarithm of HCEq/10^6^ (amastigote-equivalents/DNA equivalents of human cells).

**Figure 2 pathogens-14-00615-f002:**
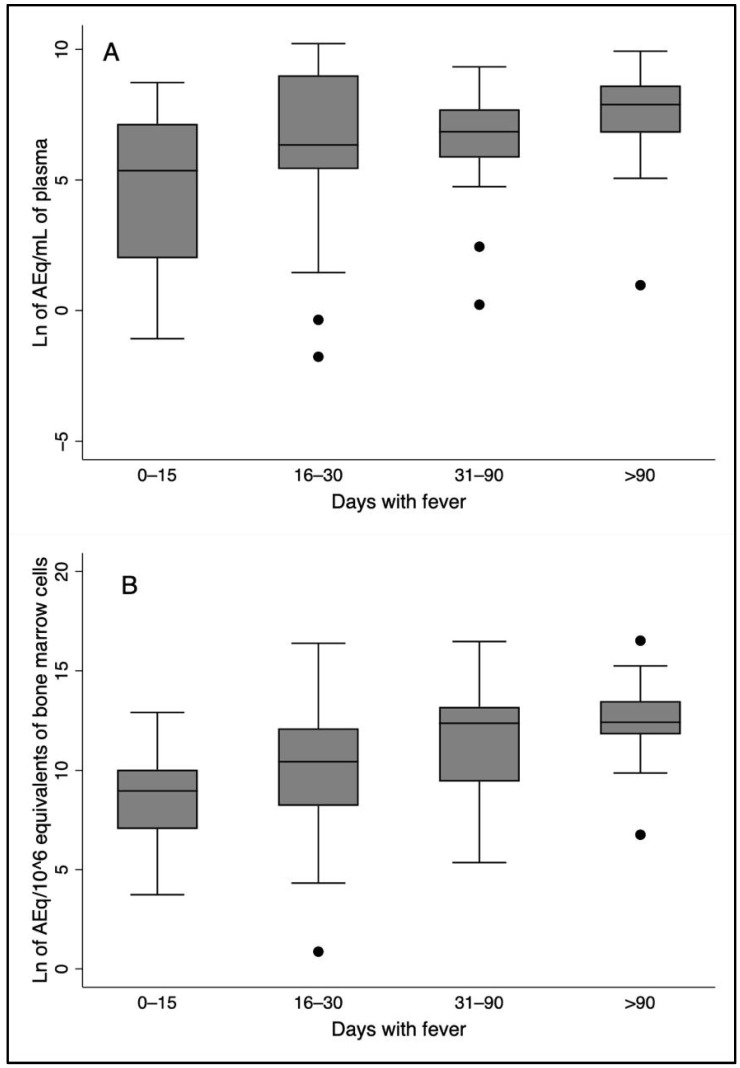
Box plots showing parasite loads increasing with time of fever both in the plasma and in the bone marrow. The correlation coefficients with time represented in the plots were (**A**) parasite load in the plasma (ρ = 0.33, *p*-value < 0.005); (**B**) Parasite load in the bone marrow (ρ = 0.38, *p*-value < 0.001).

**Table 1 pathogens-14-00615-t001:** Characteristics of the study population.

Characteristic	Number (%)	95% CI ^1^
Sex		
Male	42 (58.3)	46.1; 69.85
Female	30 (41.7)	30.2; 53.89
Age groups (Years)		
<2	17 (23.6)	14.0; 35.0
2 < 4	5 (6.9)	2.2; 15.4
4 < 15	22 (30.6)	20.2; 42.5
15 < 40	22 (30.6)	20.2; 42.5
40+	6 (8.3)	3.1; 17.3
HIV ^2^ (number, %)	13 (18.6)	10.3; 30.0
Deaths (number, %)	4 (5.6)	1.5; 13.61
Chance of death > 10% by Kala-Cal^®^	25 (34.7)	23.9; 46.9
Hemorrhages or infections	31 (43.7)	31.9; 56.0
Reported bleeding	4 (5.6%)	1.5; 13.6
Detected bleeding	15 (20.8)	12.2; 32.0
Sepsis	10 (14.1)	7.0; 24.4
Any bacterial infection	23 (31.9)	21.4; 44.0

^1^ 95% confidence interval. ^2^ Only 70 patients were tested for HIV infection.

**Table 2 pathogens-14-00615-t002:** Parasite load as measured by the concentration of kDNA and cytokines in the plasma of patients with kala-azar.

Variables	Median	Interquartile Intervals	Mean	Reference Values (Median)	Kolmogorov–Smirnov Test (*p*-Value)
Plasma kDNA (AEq ^1^/mL)	856.7	145.5–3527.9	3515.4	0 ^3^	0.000
Bone marrow kDNA (HCEq ^2^)	55.7	3.6–4008	889.9	0 ^3^	0.000
IL-1β pg/mL	0.9	0.2–2.1	2.0	0.18 (0–3.66) ^4^	0.000
IL-6 pg/mL	9.5	2.4–28.0	41.7	0 (0–0) ^4^	0.000
IL-8 pg/mL	26.2	9.8–145.5	146.9	0 (0–0) ^4^	0.000
IL-10 pg/mL	18.4	8.7–35	30.2	0 (0–0) ^4^	0.000
IL-12 pg/mL	1.2	0.0–2.5	1.8	0 (0–0) ^4^	0.000
TNF-α pg/mL	1.0	0.3–3.0	2.2	0 (0–0) ^4^	0.000
TGF-β ng/mL	23.6	11.2–42.4	39.6	NA ^5^	-

^1^ Amastigote-equivalents. ^2^ AEq/10^6^ equivalents of human cells. ^3^ Assumed to be zero in persons without visceral leishmaniasis. ^4^ Kildey et al. [24]. ^5^ Not available.

**Table 3 pathogens-14-00615-t003:** Parasite load (concentration of *Leishmania infantum* amastigotes estimated using kDNA) in the plasma of patients with kala-azar.

Markers of Severe Disease (Number of Patients)	Plasma Load (AEq) ^1^ Median, (Mean)	*p*-Value ^2^	Bone Marrow Load (AEq/10^9^HCEq) ^3^ Median, (Mean)	*p*-Value
*Age* (*years*)				
<15 (27)	508 (2513)		27 (519)	
15+ (44)	2679 (5091)	**<0.05**	137 (1494)	**<0.05**
*Sex*				
Female (30)	329 (1580)		28 (631)	
Male (41)	1452 (4898)	**<0.05**	90 (1079)	**>0.05**
*HIV*				
Yes (13)	4207 (7016)		56 (2867)	
No (57)	511 (2768)	**<0.01**	35 (500)	**>0.05**
*Hospital outcome*				
Death (4)	11,826 (11,925)		487 (605)	
Survival (67)	830 (3020)	**>0.05**	56 (907)	**>0.05**
*Chance of death > 10% according to Kala-Cal^®^*				
>10% (25)	3532 (6617)		290 (2280)	
<10% (46)	311 (1866)	**<0.001**	20 (350)	**<0.001**
*Reported bleeding*				
Yes (4)	8855 (10,440)		89 (406)	
No (67)	830 (3108)	**>0.05**	56 (919)	**>0.05**
*Detected bleeding*				
Yes (15)	823 (4427)		214 (1406)	
No (56)	876 (3276)	**>0.05**	21 (752)	**<0.05**
*Sepsis*				
Yes (10)	888 (5619)		247 (375)	
No (60)	1130 (3210)	**>0.05**	34 (988)	**>0.05**
*Any bacterial infection*				
Yes (23)	837 (3679)		425 (466)	
No (48)	1059 (3439)	**>0.05**	65 (1093)	**>0.05**

^1^ Amastigote-equivalents/mL. ^2^ Wilcoxon rank sum test. ^3^ AEq/10^9^ equivalents of human cells.

**Table 4 pathogens-14-00615-t004:** Plasma concentration of cytokines according to demographic data, and HIV-infection, and markers of kala-azar severity.

Variables (Number of Patients)	IL-1β Median, (Mean)	*p*-Value ^1^	IL-6 Median, (Mean)	*p*-Value	IL-8 Median, (Mean)	*p*-Value	IL-10 Median, (Mean)	*p*-Value	IL-12 Median, (Mean	*p*-Value	TNF-α Median, (Mean)	*p*-Value	TGF-β Median, (Mean)	*p*-Value
**Age (years)**														
<15 (27)	1.1 (2.1)		14.0 (49.3)		28.8 (130.7)		23.4 (36.4)		1.4 (2.1)		1.2 (2.5)		23.1 (40.5)	
15+ (44)	0.4 (1.9)	>0.05	7.8 (28.9)	>0.05	23.3 (174.2)	>0.05	12.8 (19.8)	*<0.05*	0.6 (1.3)	>0.05	0.9 (1.6)	>0.05	24.5 (38.0)	>0.05
**Sex**														
Female (30)	0.9 (1.9)		8.4 (55.9)		25.6 (118.7)		23.1 (32.6)		1.0 (2.2)		2.1 (3.1)		13.9 (40.6)	
Male (41)	1.0 (2.0)	>0.05	13.6 (32.6)	>0.05	26.2 (164.8)	>0.05	13.6 (28.7)	>0.05	1.3 (1.6)	>0.05	0.9 (1.7)	>0.05	28.5 (38.9)	>0.05
**HIV-infection**														
Yes (12)	1.3 (2.7)		8.1 (13.0)		46.7 (188.1)		12.4 (14.3)		1.2 (1.7)		1.6 (1.9)		16.5 (35.9)	
No (57)	0.9 (1.9)	>0.05	13.0 (48.8)	>0.05	25.4 (140.5)	>0.05	21.8 (34.2)	*<0.05*	1.3 (1.9)	>0.05	1.0 (2.3)	>0.05	24.05 (40.1)	>0.05
**Hospital outcome**														
Death (4)	1.3 (1.3)		24.7 (44.4)		48.1 (173.9)		12.0 (31.5)		1.4 (1.8)		1.2 (1.8)		18.6 (20.4)	
Survival (62)	0.9 (2.1)	>0.05	9.1 (42.2)	>0.05	26.4 (147.5)	>0.05	18.7 (30.5)	>0.05	1.2 (1.8)	0.61	1.0 (2.3)	>0.05	24.1 (20.4)	>0.05
**Chance of death ^2^**														
>10% (23)	1.0 (2.5)		15.4 (44.3)		65.2 (206.5)		16.2 (27.8)		1.1 (1.6)		1.0 (2.3)		24.5 (45.4)	
<10% (43)	0.9 (1.8)	>0.05	9.5 (44.3)	>0.05	24.6 (118.5)	>0.05	19.1 (32.0)	>0.05	1.3 (1.9)	>0.05	0.9 (2.2)	>0.05	22.3 (36.1)	>0.05
**Reported bleeding**														
Yes (3)	1.6 (1.8)		34.1 (63.7)		78.6 (246.1)		34.7 (51.3)		2.0 (2.1)		0.4 (1.8)		15.7 (19.8)	
No (63)	0.9 (2.0)	>0.05	8.6 (41.2)	*>0.05*	24.9 (144.5)	>0.05	16.6 (29.5)	>0.05	1.2 (1.8)	>0.05	1.0 (2.3)	>0.05	23.6 (40.3)	>0.05
**Detected bleeding**														
Yes (13)	0.9 (2.8)		17.2 (24.9)		88.5 (242.7)		16.1 (24.1)		2.0 (2.5)		2.1 (2.2)		42.1 (59.8)	
No (53)	1.0 (1.9)	>0.05	8.6 (46.6)	>0.05	24.6 (126.2)	*>0.05*	21.8 (32.1)	>0.05	1.2 (1.7)	>0.05	1.0 (2.3)	>0.05	20.9 (34.3)	>0.05
**Sepsis**														
Yes (10)	1.6 (1.9)		23.0 (137.3)		110.2 (229.9)		36.9 (37.3)		1.9 (2.5)		2.4 (2.5)		18.3 (31.9)	
No (56)	0.8 (2.0)	>0.05	8.1 (25.3)	>0.05	24.7 (134.7)	>0.05	16.1 (29.3)	>0.05	0.9 (1.7)	*<0.05*	0.9 (2.2)	>0.05	25.7 (40.7)	>0.05
**Any bacterial infection**														
Yes (22)	1.6 (2.4)		25.2 (83.1)		67.0 (203.9)		30.5 (40.6)		1.9 (2.1)		2.4 (2.8)		18.9 (33.0)	
No (44)	0.7 (1.8)	*<0.05*	7.2 (21.9)	*<0.05*	22.6 (121.8)	*<0.05*	16.0 (25.5)	*<0.05*	0.9 (1.7)	>0.05	0.8 (2.0)	*<0.05*	29.5 (42.5)	>0.05

^1^ Wilcoxon ranksum test. ^2^ Kala-Cal^®^.

## Data Availability

Data is available through permission given directly by the authors IS and CC.

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
