# Peer review of "Immune–Pathological Correlates of Disease Severity in New-World Kala-Azar: The Role of Parasite Load and Cytokine Profiles"

_pathogens, 2025, doi:10.3390/pathogens14070615_

Round 1
Reviewer 1 Report
Comments and Suggestions for Authors
Dear authors,
The entry manuscript (MS) “Immune-Pathological Correlates of Disease Severity in New World Kala-Azar: The Role of Parasite Load and Cytokine Profiles” fits very well in the in the scope of Pathogens MDPI. After a detailed evaluation of the manuscript, my major suggestion comprises minor editorial corrections. Therefore, please solve small points before the final acceptance:
- The keywords should provide an alternative way to find studies from similar topics. Therefore, the addition of distinct words from the title could help other researches to find your manuscript. Please, consider providing just keywords not presented in the title;
- Please insert the symbol “γ” from IFN-γ instead of “g”, as well the symbols “α” from TNF- α and “β” from IL-1β in the main text;
- There are some changes in the text font during the MS. Please, adjust the text font in lines 77-78, 101-103 and 144-146;
- infantum in line 319 must be in italic;
- What is IQ in line 194? Readers could be not necessarily familiar with the analysis made in this MS. Please, consider inserting at least the description the full name of IQ;
- In table 2, should the comparison with health donors not be made? How much are the reference values from health individuals? Other pathologies were removed from the analysis?
- Should Figure 3 not be a linear regression curve? The presented image is not necessary a figure. It´s most like a table. It´s better to adequate to a curve with the data or change to a table.
Please, consider review minor editorial corrections, such as font standardization and insertion of symbols in cytokines’ names to improve the MS.
Author Response
|
Dear Editor and reviewer 1,
|
|
|
|
Thank you very much for taking the time to review this manuscript. Please find the detailed responses below and the corresponding revisions/corrections highlighted/in track changes in the re-submitted files. The tracking was made through highlighting in yellow words and phrases changes, not by the usual Word tracking system because we were afraid of dislocating the referred line numbers. Therefore, small changes, such as fonts, p-values, Greek characters for cytokines, and isolated words may not be identified in the tracked versions. Due to comments of reviewer 2, we added three new cytokines regarding the role of IL-27, including its role in spleen white pulp disruption. To make the issue more complete, we had to add another reference. Best regards, Carlos H N Costa Corresponding author
|
||
Pathogens Reviewers Reports
Reviewer 1.
The entry manuscript (MS) “Immune-Pathological Correlates of Disease Severity in New World Kala-Azar: The Role of Parasite Load and Cytokine Profiles” fits very well in the in the scope of Pathogens MDPI. After a detailed evaluation of the manuscript, my major suggestion comprises minor editorial corrections. Therefore, please solve small points before the final acceptance:
Comment 1: The keywords should provide an alternative way to find studies from similar topics. Therefore, the addition of distinct words from the title could help other researches to find your manuscript. Please, consider providing just keywords not presented in the title;
Response 1: Thanks for the wise suggestion! We did so.
Comment 2: Please insert the symbol “γ” from IFN-γ instead of “g”, as well the symbols “α” from TNF- α and “β” from IL-1β in the main text;
Response 2: Job done! Thanks!
Comment 3: There are some changes in the text font during the MS. Please, adjust the text font in lines 77-78, 101-103 and 144-146;
Response 3: Text fonts adjusted.
Comment 4: infantum in line 319 must be in italic;
Response 4: Job done.
Comment 5: What is IQ in line 194? Readers could be not necessarily familiar with the analysis made in this MS. Please, consider inserting at least the description the full name of IQ;
Response 5: Interquartile interval. Corrected accordingly.
Comment 6: In table 2, should the comparison with health donors not be made? How much are the reference values from health individuals? Other pathologies were removed from the analysis?
Response 6: The concentration of cytokines changes accordingly to the method, e.g., flow cytometry versus immunoassay. We found this single study quantifying most cytokines together through the same method we had done. Then we took the reported values as normal values, since the donors were healthy blood donors.
Comment 7: Should Figure 3 not be a linear regression curve? The presented image is not necessary a figure. It´s most like a table. It´s better to adequate to a curve with the data or change to a table.
Response 7: Theoretically it is a regression of dependent multiple values to a straight line, although in a hyperplane. However, we think the comment is right, and it may not be appropriate to publish the raw data of a multivariate outcome in a scientific journal, and we decided to insert the figure as supplemental material instead.
Comments on the Quality of English Language
Comment 8: Please, consider review minor editorial corrections, such as font standardization and insertion of symbols in cytokines’ names to improve the MS.
Response 8: Thanks again. We did this task.

Reviewer 2 Report
Comments and Suggestions for Authors
This study presents a novel approach to “Immune-Pathological Correlates of Disease Severity in New World Kala-Azar: The Role of Parasite Load and Cytokine Profiles.” The author has justified their study in multiple approaches with in-depth supportive literature. This study could be beneficial for many researchers. However, further discussion on a practical application basis could enhance the strength of your paper:
Major suggestions:
- Highlight more by adding related data on including INF-Y and IL-27 in the analysis to better characterize TH-1-mediated and regulatory immune response, which are central to kala-azar pathogenesis. Also, please clarify why you exclude these parameters.
- Please highlight why you did not perform longitudinal cytokine monitoring in representative patients to capture dynamic changes in immune responses throughout disease progression and treatments. In addition, please clarify in not analyze functional immune response.
- Please incorporate the parasite genotype (strain and genomic variability) at the different stages of disease and with the treatments.
- Please make more clarification in the multiple testing correction, and indicate which associations remain statistically significant after applying appropriate adjustments in all the data.
- Please incorporate the flow chart of the method sections (including in brief, patient selection, inclusion, Exclusion, and other study). This can improve the significance and easy understanding of your study.
Minor suggestions
- Clarify terminology at first and use it all over a manuscript (eg. Abstract HCEq, AEq, CBA)
- Standardize statistical legends (eg. P= 0.05 vs P < 0.05)
- Extend the figure caption and make it more informative.
- Typological error (eg. 200.0 µL vs 200 µL).
- Error in table “Table 3. Table2”
- In the description of the table, the reference and abbreviation are confusing, please differentiate them to easily identify.
Author Response
Pathogens Reviewers Report
Reviewer 2
This study presents a novel approach to “Immune-Pathological Correlates of Disease Severity in New World Kala-Azar: The Role of Parasite Load and Cytokine Profiles.” The author has justified their study in multiple approaches with in-depth supportive literature. This study could be beneficial for many researchers. However, further discussion on a practical application basis could enhance the strength of your paper:
Major suggestions:
Comment 1. Highlight more by adding related data on including INF-Y and IL-27 in the analysis to better characterize TH-1-mediated and regulatory immune response, which are central to kala-azar pathogenesis. Also, please clarify why you exclude these parameters.
Response 1: We are sorry, but we just missed the paper from Ansari eta, 2011 and the importance of IL-27. We added it and two more references to make the suggested relevance of IL-27 clearer (Zhang et al, 2014 and Montes de Oca et al, 2020). After reading them, we changed the wording from lines 396-397, and 400 and added the word “stronger” in line 404. We also added a phrasing at the end of the penultimate paragraph, adding the importance of IL-27 for the disruption of spleen architecture and the final outcome of severe kala-azar on lines 489-483.
Comment 2: Please highlight why you did not perform longitudinal cytokine monitoring in representative patients to capture dynamic changes in immune responses throughout disease progression and treatments. In addition, please clarify in not analyze functional immune response.
Response 2: We agree that it would really be important, but we had not the sufficient funding for extending the analyses.
Comment 3: Please incorporate the parasite genotype (strain and genomic variability) at the different stages of disease and with the treatments.
Response 3: Sorry, but we do not have this information.
Comment 4: Please make more clarification in the multiple testing correction, and indicate which associations remain statistically significant after applying appropriate adjustments in all the data.
Response 4: Bonferroni’s test aims to reduce the chance of type I error, e.g., stating that an association exists when it does not. The test is performed by multiplying the number of tests by the p-value threshold (0.05). This means that we would be 0.05 x 42 (at least) < 0.001. No test reached such a threshold for statistical significance for cytokine testing.
Comment 5: Please incorporate the flow chart of the method sections (including in brief, patient selection, inclusion, Exclusion, and other study). This can improve the significance and easy understanding of your study.
Response 5: We may be missing this point. We tried some imagery, as suggested, but they seemed too confusing. Therefore, we decided to write again the whole paragraph with the clinical approaches. Hence, we kindly ask your permission not to show any image.
Minor suggestions
Comment 6: Clarify terminology at first and use it all over a manuscript (eg. Abstract HCEq, AEq, CBA)
Response 6: We think we improved the parasite count units, but we did not identify any issue regarding CBA.
Comment 7: Standardize statistical legends (eg. P= 0.05 vs P < 0.05)
Response 7: Job done.
Comment 8: Extend the figure caption and make it more informative.
Response 8: Job done. Thanks again.
Comment 9: Typological error (eg. 200.0 µL vs 200 µL).
Resposne 9: Corrected.
Comment 10: Error in table “Table 3. Table2”
Response 10: Corrected.
Comment 11: In the description of the table, the reference and abbreviation are confusing, please differentiate them to easily identify.
Response 11: We improved the description of the table; the reference and abbreviation and we suppose they are fine now.
